# Why Do Dietary Flavonoids Have a Promising Effect as Enhancers of Anthracyclines? Hydroxyl Substituents, Bioavailability and Biological Activity

**DOI:** 10.3390/ijms24010391

**Published:** 2022-12-26

**Authors:** Aleksandra Golonko, Adam Jan Olichwier, Renata Swislocka, Lukasz Szczerbinski, Włodzimierz Lewandowski

**Affiliations:** 1Clinical Research Centre, Medical University of Bialystok, 15-276 Bialystok, Poland; 2Prof. Wacław Dąbrowski Institute of Agricultural and Food Biotechnology—State Research Institute, 02-532 Warsaw, Poland; 3Department of Chemistry, Biology and Biotechnology, Bialystok University of Technology, 15-351 Bialystok, Poland; 4Department of Endocrinology, Diabetology and Internal Medicine, Medical University of Bialystok, 15-276 Bialystok, Poland

**Keywords:** synergistic effects, quercetin, GSH-depletion, multi-drug resistance, structure-activity relationship

## Abstract

Anthracyclines currently play a key role in the treatment of many cancers, but the limiting factor of their use is the widespread phenomenon of drug resistance and untargeted toxicity. Flavonoids have pleiotropic, beneficial effects on human health that, apart from antioxidant activity, are currently considered small molecules—starting structures for drug development and enhancers of conventional therapeutics. This paper is a review of the current and most important data on the participation of a selected series of flavonoids: chrysin, apigenin, kaempferol, quercetin and myricetin, which differ in the presence of an additional hydroxyl group, in the formation of a synergistic effect with anthracycline antibiotics. The review includes a characterization of the mechanism of action of flavonoids, as well as insight into the physicochemical parameters determining their bioavailability in vitro. The crosstalk between flavonoids and the molecular activity of anthracyclines discussed in the article covers the most important common areas of action, such as (1) disruption of DNA integrity (genotoxic effect), (2) modulation of antioxidant response pathways, and (3) inhibition of the activity of membrane proteins responsible for the active transport of drugs and xenobiotics. The increase in knowledge about the relationship between the molecular structure of flavonoids and their biological effect makes it possible to more effectively search for derivatives with a synergistic effect with anthracyclines and to develop better therapeutic strategies in the treatment of cancer.

## 1. Introduction

Cancer is now one of the leading causes of death worldwide. According to the World Health Organization, in 2020, 10 million people died of cancer worldwide, and according to the American Cancer Society, in 2022, 1,918,030 new cases will be detected in the United States, of which over 609,360 people will die of cancer [1]. Due to the growing number of cancers, there is a constant increase in the need for the development of new, more effective and comprehensive therapeutic approaches. The new approach to treatment also includes combination therapies involving the administration of substances with different mechanisms of action, which reduces the likelihood of developing drug-resistant cells. In addition, the use of both therapeutics in the optimal dose reduces the risk of side effects [2].

Anthracycline drugs, which are among the most effective anticancer drugs, are on the list of essential medicines of the World Health Organization [3] and are used in the treatment of cancers such as breast cancer, lymphoma, sarcoma and many others [4]. The most frequently used anthracillins in oncology are doxorubicin, daunorubicin, epirubicin and idarubicin, whose mechanism of action is multifactorial, including disruption of DNA integrity, binding to the cell membrane and induction of oxidative stress by the intensive generation of free radicals [5]. Unfortunately, the high anticancer activity of this group of potent drugs is associated with a long list of adverse effects in the form of genotoxicity resulting in the development of other cancers and cardiotoxicity, as well as the risk of multidrug resistance (MDR). To date, many compounds have been studied to correct for the side effects of anthracyclines and to prevent the occurrence of MDR, among others, plant secondary metabolites, which include, among others, phenolic acids, terpenoids and flavonoids that are widely distributed in nature and can interact with single or multiple targets [6].

Flavonoids with high antioxidant activity and the ability to interact with membrane transporter proteins involved in the development of MDR constitute a promising group of chemical compounds to be considered in the context of potential use in combination therapy with anthracycline drugs. In addition, this broad group of compounds has a proven chemopreventive effect, among others, in the development of estrogen-related [7], gynecological [8] and stomach and colorectal cancers [9].

According to the Google Patents database, more than 16,000 patents containing the phrase “flavonoid anti-cancer” have been published in the last five years (since 2017). It seems that the number of patent applications describing the anticancer effects of phenolic compounds and their derivatives is much higher, as indicated by the ever-increasing number of literary reports on the subject—more than 23 thousand publications on the current state. The interest of the scientific community in the combination of flavonoids with chemotherapeutic drugs is constantly growing—by 2015, about 170 articles covering the words “flavonoid & anthracycline” had been published, and in the following 5 years there are already about 340 publications in the PubMed database.

The growing trend in this topic is due to previous observations that have yielded promising therapeutic results. This subject is currently being explored using more accurate experimental methods and “omics” approaches [10]. Integrated genomics [11], epigenomics [12] and proteomic [13] and metabolomic data [14] using bioinformatics analysis methods have greatly accelerated the drug discovery process [15]. Indeed, flavonoids are no longer just examined as plant metabolites with health-promoting properties, but also as precursor molecules for drug design [16]. With extensive access to large (and constantly expanding) datasets used in in silico methods with artificial intelligence tools, studies of synergistic effects involving a range of complex intermolecular interactions are proceeding much more rapidly [17]. Such a comprehensive approach, however, requires the selection of precise methods for data analysis and the integration of many techniques used in bioorganic chemistry, structural chemistry, molecular biology, genetics, histology and biochemistry. The use of this approach makes it possible to select compounds with the best drug-enhancing potential, as well as to characterize the structural elements responsible for the observed biological effects [18,19].

It is assumed that hydroxyl groups forming structural systems in ortho, meta or para positions in the case of flavonoids are important in reducing free radicals and reducing cell viability [20]. Hydroxyl groups are also crucial for interactions (non-covalent, electrostatic, H-bonding) with other compounds and biological macromolecules—cell membrane, proteins, DNA [21]. Therefore, an integrated approach to the search for structures with the highest possible synergistic effect with anthracycline drugs requires a thorough analysis of a selected group from a wide list of flavonoids that differ from each other by one functional group. Flavonoids are often studied by subclasses with the assumption that structures within one subclass behave similarly, however, it should be emphasized that slight structural differences related to the number and position of hydroxyl groups are associated with significant differences in biological activity [22]. 

The aim of this review is to present and evaluate the mechanisms behind the synergistic effects of phenolic compounds from the group of flavonoids and anthracycline drugs. We have presented the mechanisms of synergistic activity of a series of logical series of ligands with an increasing number of hydroxyl groups: chrysin (flavone), apigenin (flavone), quercetin (flavonol), kaempferol (flavonol) and myricetin (flavonol) at the three most important levels of molecular activity common to the pathways of toxicity induced by anthracycline antibiotics (1) prooxidative activity, (2) interaction with membrane proteins; and (3) genotoxicity.

This review includes literature reports relevant to the development of scientific knowledge available in the PubChem, Medline, Scopus, Web of Science and Google Scholar databases containing the phrases “flavonoid AND anthracycline OR doxorubicin OR daunrubicin OR idarubicin”. A selective search was also used for each of the flavonoids using their common names contained in the presented figures and tables. The available search results were narrowed down to the last five years, and for interaction descriptions, the search period was extended to the last seven years using keywords corresponding to a specific molecular target: DNA, membrane receptors or intracellular proteins. Pharmacokinetic properties and bioavailability are described based on data including the keywords “pharmacokinetics AND flavonoids OR [common name of flavonoid]”.

## 2. Anthracyclines—Toxicity and Mechanism of Action

The mechanism of action of anthracycline drugs, also known as quinone-chelators, is very complex, and each drug belonging to this group (doxorubicin, daunorubicin, epirubicin, idarubicin, mitoxantrone, valrubicin) is structurally different, making its use dedicated to specific types of cancer-solid tumors and hematological malignancies [23]. These drugs are structurally very similar but differ in lipophilicity and membrane permeability. Epirubicin is a stereoisomeric derivative of doxorubicin with a longer half-life, while idarubicin, a daunorubicin derivative (without the methoxy group), has a higher lipophilicity and higher cellular uptake than daunorubicin [24].

The general mechanism of anthracycline activity is based on several major mechanisms including (i) formation of intercalation complexes with DNA, covalent bonds and base modifications [25] (ii) inhibition of topoisomerase II activity [26], resulting in disruption of DNA repair, replication and transcription; (iii) pro-oxidant activity [27]—generation of free radicals, lipid peroxidation and oxidative damage to DNA and induction of apoptosis via the mitochondrial pathway [28]. Anthracyclines belong to quinone compounds, termed “notorious redox cyclers”[29]. Their pro-oxidant activity, and thus redox transformations, generating free radicals by oxidation to unstable semiquinones, will be key to cytotoxicity [30]. 

Although anthracyclines are effective cytostatics, their main disadvantage is the occurrence of systemic toxicity, including cardiotoxicity, gonadotoxicity and MDR [31,32,33,34,35]. Resistance to anthracyclines can be natural in some cells without exposure to chemotherapy or can be acquired after drug administration by stimulating efflux and inhibiting drug influx into cells due to the activity and expression of ABC membrane proteins, including P-glycoprotein (P-gp) [36]. Currently, it is recognized that the main role in multi-drug resistance of cancer cells in vitro is played by ABCB1 (P-gp), ABCC1 (MRP1), ABCC2 (MRP2), ABCC4 (MRP4), ABCG2 (BCRP) and lung resistance protein (LRP) [37]. The expression and activity of the transporters are under the complex control of redox signals, which are activated in response to the chemotherapeutic agents [38]. Interestingly, the upregulation of one type of efflux pump is associated with the development of pleiotropic resistance to a number of non-related anticancer drugs [39], and in addition, cross-resistance to many other pharmaceuticals of varying structure [40]. The concept of inhibition of transporter proteins is obviously very logical, but potential suppressors (especially of herbal origin) can easily interact with efflux pathways (transport proteins—xenobiotic efflux) and metabolism. Anthracyclines are substrates of cytochrome P450 2D6 and 3A4 (CYP2D6 and CYP3A4), so reducing drug clearance by inhibitor administration increases the risk of adverse effects, including cardiotoxicity [41].

Many research teams have taken up the topic of modifying the DOX structure in order to enhance cytotoxic properties, e.g., by creating metal complexes and derivatives with a different form of delivery [42,43]. In addition, another promising approach in the search for more effective therapeutic methods based on anthracyclines is the parallel use of several pharmaceuticals with an additive and/or synergistic effect [44,45]. Adjuvant therapies using conventional drugs as well as natural, bioactive phenolic compounds of plant origin may potentially enhance the cytotoxic effects of anthracyclines. At the same time, they can mitigate the negative effect on the functioning of cardiomyocytes, which are the cells most exposed to damage during anthracycline therapy [46].

In order to overcome limitations associated with the use of anthracycline drugs, such as short retention time, systemic toxicity and the risk of drug resistance, efforts are being made to increase their effectiveness and minimize the risk of side effects [47]. The key issue is to develop such a therapeutic strategy in which the synergistic effect with the anthracycline drug and the reduction of non-targeted toxicity would take place through different molecular pathways/mechanisms. Then the reduction of untargeted toxicity would not reduce the therapeutic effectiveness of the drug [35].

## 3. Flavonoids—Structure, Bioavailability and Molecular Activity

Flavonoids are a large class of polyphenolic compounds that structure is based on a 15-carbon C6-C3-C6 structural skeleton, where the A and B rings have a phenolic form, and a C heterocyclic pyrane ring (Figure 1). Flavonoids are widespread in the plant world [48] and are therefore integral components of the human diet [49]. They occur naturally in the form of aglycones, glycosides and methylated derivatives [50]. The average intake of flavonoids among humans ranges from 200 mg to 1 g per day [51] although current sources report an average intake of 356 mg/day among men and 328 mg/day among women [52]. In addition to natural dietary sources, flavonoids (as pure aglycones or plant extracts) are also a component of dietary supplements in doses far exceeding those provided with food products [53].

The beneficial, health-promoting effect of flavonoids is supported by many studies, which have proven a positive, multi-directional effect of flavonoid intake on the improvement of health parameters, including cognitive functions [54], biochemical parameters in the metabolic syndrome [55], reducing the risk of cancer [7].

Their pleiotropic effect results from the ability to interact with many cellular targets and antioxidant properties, thus limiting the negative effects of oxidative stress accompanying inflammation and cancer (Table A1—cytotoxic effect of flavonoids in in vitro models). The broad spectrum of flavonoid biological activity, including many signaling pathways, leads to considering this group of compounds as structures with both chemopreventive and therapeutic properties [56]. Therefore, the antioxidant activity of this group of compounds will have a beneficial effect on the comprehensive approach to the prevention and treatment of diseases with an immune, metabolic, viral and microbiological background [57,58]. Recognition of the biological activity of flavonoids and the ability to modulate the catalytic activity of cytochromes and transporters made it possible to formulate recommendations regarding the consumption of certain groups of products during conventional pharmacotherapy. Among other things, it has been observed that licorice (*Glycyrrhiza* sp.), which is a rich source of phytochemicals and bioactive compounds, including flavonoids, is potentially dangerous due to its ability to inhibit drug-metabolizing cytochrome P-450. Direct action as well as modulation of the pharmacokinetics and pharmacodynamics of drugs may result in increased potential for chronic and acute drug toxicity imposed by substrate drugs [59]. Therefore, a detailed understanding of the ways, targets and mechanisms of interaction of flavonoids with drugs, including anti-cancer drugs, may increase the scope of therapeutic safety and contribute to finding beneficial clinical applications in this field [59].

### 3.1. General Structure of Flavonoids

Most of the available literature data relate to the description of the properties of single structures, without tracking structure–activity or structure–toxicity relationships, in logical, structural series of phenolic compounds [60,61,62]. Therefore, a review of physicochemical properties among an ordered structural group of compounds makes it possible to systematize knowledge about the influence of substituents on the formation of a biological effect in in vitro and in vivo conditions [63,64]. The distinguishing feature of flavonoids from other bioactive compounds of plant origin is the high delocalization of the electron charge from the A and B rings through the C ring and the presence of a double bond in the C2=C3 position [65] (Figure 1—flavone backbone)**.** This binding makes the compounds more planar, which results in elongation of the conjugated system in the molecule, which provides them with beneficial antioxidant properties [66], but also the ability of this group of compounds to bind to hydrophobic regions of proteins, including enzymes involved in the regulation of cellular metabolic pathways [67,68].

Among the flavonoids analyzed in this review, all have an unsaturated C2=C3 bond, however, the review undertaken covers a logical series of flavonoids starting from chrysin, which does not have any substituents in the B ring, through apigenin with an additional hydroxyl group in the C4′ position, to kaempferol → myricetin with a C3-OH moiety and an increasing number of –OH groups in ring B in position C3′ and C5′, respectively (Figure 1). It is known that the 3-OH hydroxyl moiety is important for conferring anti-radical properties, and the catechol system (C3′, C4′) found in quercetin (Figure 1) will be responsible for significantly higher free radical scavenging capabilities compared to other flavonoids [69]. However, the acceptor-donor properties of flavonoids do not fully explain their full spectrum of activity observed in complex biological systems such as human cells (Table A1—cytotoxic effect of flavonoids in in vitro models) and tissues, where exogenous compounds are subject to complex metabolism and series of interactions depending on the reaction environment [70,71]. Therefore, a systematic review of a selected series of ligands provides the opportunity to compare the biological properties described so far in relation to the precisely characterized chemical and electronic structure [72,73].

### 3.2. Absorption and Metabolism of Flavonoids Determine In Vivo Activity

Promising results of the anticancer activity of flavonoids obtained in in vitro experiments prompt to undertake actions toward their therapeutic clinical use. Previously published results based on in vitro experiments should be treated with great caution when the studies tested aglycones or forms conjugated with the sugar moiety, and not active phase II metabolites, and the concentrations used in in vitro tests significantly exceeded the nanomolar ranges that naturally reach the plasma. It is believed that the aglycon form of flavonoids can be absorbed by passive diffusion, although it is possible that some glycosides can also penetrate the intestinal wall, but at a slower pace [74,75]. Bioavailability will therefore depend on the individual characteristics of the structure and expression of membrane transporters [72].

It is known that intensive metabolism of flavonoids in enterocytes and hepatocytes by UGT (UDP-glucuronosyltransferases) and SULT (sul-fotransferases) enzymes leads to the formation of glucuronides and sulfates, then excreted by ABC (ATP-binding cassette) transports (MPR2—multidrug resistance-associated protein 2) or BCRP (breast cancer resistance protein). Thus, they act as chemical regulators (inducers or inhibitors) of the expression of phase II enzymes and membrane transporters. Inhibition of the activity of OATP (organic anion transporting polypeptide), BCRP or MPR2 transporters will depend on the chemical form of the flavonoid and this property in many cases will be closely related to the structure and individual for a given compound. A modification in the form of a change in one of the substituents can dramatically change the affinity of a flavonoid to a given transporter. A large difference in the effect on the MPR2 transporter was observed, among others, for chrysin and chrysin sulfate, the IC50 value against MPR2 is above 100 μM, and the chrysin glucuronide has an IC50 = 11.2 μM [72].

Even if the flavonoid does not interact with the membrane transporter, there are many conditions that will dictate their penetration into the cell. The passive diffusion path of flavonoids is conditioned by many complex physicochemical descriptors, e.g., the atomic charge of the molecule or the solvation energy. So ultimately, the absorption of flavonoid molecules will be influenced by the size of the molecule, configuration, lipophilicity, solubility and dissociation constant, which are dictated by the electron charge distribution. Individual flavonoids will also show completely different levels of intracellular accumulation depending on the cell line [74,75], flavonoid class [76] and spatial structure [77]. Even structural factors such as the number of hydroxyl groups can determine the efficiency of transport across biological membranes. In this case, more hydrophobic polyphenols with more hydroxyl groups can penetrate deeper into lipid bilayers, as even the presence of one hydroxyl group can dramatically change the affinity of flavonoids for membrane lipids [78]. For example, ring A of chrysin will be facing the aqueous phase of the lipid bilayer, and ring B for myricetin [79]. In the case of cell membrane permeability, it seems that both quercetin and kaempferol can be efficiently transported inside the cell, and naringenin (4,5,7-trihydroxy flavanone), which, unlike the other two flavonoids, does not have hydroxyl groups in ring B, will not penetrate into the cell [78].

However, low permeability does not exclude a biological effect that will take place within the cell membrane or toward membrane transporters/receptors [80]. Indeed, thanks to the use of cytometric methods, it has been observed that quercetin can pass into the cell and locate in the cell membrane [74], but reports on the influence of a flavonoid, e.g., mitochondrial biogenesis, do not clearly indicate the possibility of transporting this compound to this organelle [81]. The subject of membrane proteins is important not only when considering the aspects of bioavailability of xenobiotic compounds, but also the complex phenomenon of multiple drug resistance (MDR), which is responsible for the failure of chemotherapy due to overexpression of transport proteins [82]. 

Very low bioavailability (due, among other things, to low solubility) is one of the main problems and limitations in the use of flavonoids as potential anticancer therapeutic drugs in clinical trials, since oral intake of high doses of flavonoids does not guarantee such concentrations in the target tissue. It has been estimated that oral delivery of as much as 400 mg of chrysin—one of the discussed flavonoids in this review—leads to its appearance in plasma in only trace amounts (0.003–0.02% bioavailability) [83], while in the case of quercetin, no plasma concentrations were determined after oral ingestion of 4 g of the aglycone [84]. Due to relatively low bioavailability, work is underway on other delivery systems, e.g., Quercetin Phytosome^®^ (a modified form of quercetin) with improved solubility in vitro, oral absorption and plasma concentrations 20 times higher than standard quercetin [85].

### 3.3. Bioavailability of Flavonoids Conditioned by Physicochemical Parameters

Bioavailability parameters determined by means of physicochemical descriptors will enable comparison and prediction of the fate of a chemical compound in the systemic environment, as well as the possible mechanism of transport and accumulation in a living organism. Knowledge of parameters such as solubility, lipophilicity, distribution of electron charge in the entire molecule and energy of HOMO (Highest Occupied Molecular Orbital) and LUMO (Lowest Unoccupied Molecular Orbital) boundary orbitals. All these parameters allow for an overall assessment of the so-called ADME (absorption, distribution, metabolism, and elimination) in biological systems. Therefore, flavonoids as starting structures for the synthesis of new potential drugs or additive compounds should be considered in terms of the electron structure, the distribution of the electron charge and the relocation of this charge upon interaction with a macromolecule or a metal ion present in the cells microenvironment. By using molecular modeling, it is possible to predict the most stable compound conformation, solubility and reactivity of the molecule. It is known that the energies of the HOMO and LUMO boundary orbitals are important for intermolecular interactions, and the molecules with the smallest HOMO-LUMO gap (difference between HOMO and LUMO) will be characterized by enhanced interaction and often a greater affinity for cancer cells [86]. Currently, there is increasing use of predictive tools for determining octanol/water partition coefficients (logP) and water solubility (logS), as well as complex absorption and permeability descriptors, obtained based on machine learning and artificial intelligence methods [87]. These methods make it possible to define important features of the structure of flavonoids that are crucial for their bioavailability and biological activity, including anticancer [88,89].

The calculated result of LogS showed that chrysin is the least soluble molecule, but all of them have optimal lipophilicity between −0.4 to 5.6 (Ghose’s rules) [39], but lipophilicity of the ionizable groups at pH 7.4, called LogD, is much more critical for physiological absorption [90]. Low LogP values obtained for apigenin and chrysin are associated with a small number of hydroxyl groups. Myricetin is the only one of the analyzed compounds to have six hydrogen bond donors, which violates Lipinski’s rule of five (number of hydrogen bond acceptors must be NHBD > 5) [91] for rational drug design for oral administration. An equally important parameter is the molecular polar surface area (PSA), i.e., the surface belonging to polar atoms that correlate with passive transport through membranes. The computation of PSA based on the summation and divisions of the surface polar fragments is called topological PSA (TPSA) [92]. There is an inverse relationship between absorption and the topological polar surface area (TPSA) parameter, which may indicate that the myricetin with the highest value will be the worst absorbed. High value (TPSA > 140 Å^2^) is associated with poor permeability through cell membranes. Moreover, flavonoids with a higher LogP value and a lower TPSA value have the ability to cross the blood-brain barrier with potential effects [93]. Depending on the purpose for which a given flavonoid is to act, other physicochemical and structural parameters will be important. For inhibition of CYP3A4 activity, molecules with lower lipophilicity and low molecular weight will be preferred, while inhibition of OATP will be more effective in molecules with higher lipophilicity and higher TPSA [94].

There is a method for determining favorable properties for absorption from the gastrointestinal tract from two calculated descriptors: WLOGP (WLOGP—one of the LogP prediction method [95].) and TPSA, which was developed from a dataset of known molecules with good and poor absorption [96]. It is presented in the form of an ellipse, the so-called BOILED-egg (Brain Or IntestinaL EstimateD permeation method) [96]. A high probability of passive absorption from the gastrointestinal tract is represented by a white area (Figure 2). The yellow region represents a high probability of crossing the blood-brain barrier (BBB). Red dots indicate that the particles are actively effluxed by P-gp (all of the analyzed flavonoids). The analysis of the distribution of the analyzed flavonoids in the WLOGP (Y) and TPSA (X) axes shows that chrysin is the only flavonoid among the entire series of ligands that is characterized by the value of these two parameters enabling penetration through the BBB barrier. The remaining compounds, except for myricetin, have WLOGP and TPSA values enabling passive absorption from the gastrointestinal tract. Graphically, there is a clear downward trend in absorption in the direction of chrysin → myricetin, which emphasizes the fact of limited bioavailability due to the greater number of hydroxyl groups in the flavonoid molecule.

Understanding the detailed structure and distribution of the electron charge determining the acceptor-donor and antioxidant properties of flavonoids is possible thanks to the tools of computational chemistry [65]. Such descriptors based on Density Functional Theory (DFT) are currently among the many types of electrochemical parameters taken into account in the selection and design of macromolecule inhibitors, e.g., P-gp [97] or other transport proteins of this class (ABC) conferring a drug-resistant phenotype in tumor cells [98]. The analysis of the obtained theoretical parameters made it possible to observe, among others, that the solvation energy and the dipole moment will be positively correlated with the inhibition of P-gp [99], so these parameters should be taken into account when designing specific inhibitors of this transporter.
Figure 2Overview of the BOILED-Egg construction of analyzed flavonoids. Obtained using Swiss ADME website [100] based on lipophilicity (WLOGP) and polarity (TPSA [Å^2^]) computation.
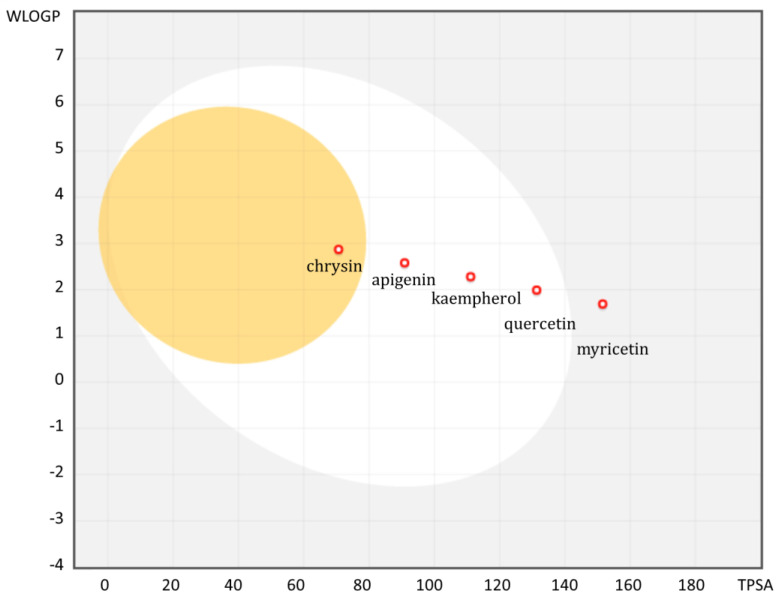



Table 1 presents the estimation of selected physicochemical parameters for the analyzed ligand series, such as solubility, lipophilicity and TPSA, which should be taken into account depending on the specific molecular target and the possible transport route of the potential pharmaceutical. Data on physicochemical properties, such as SILICOS-IT LogP and ESOL LogS, were obtained on the basis of the SwissADME prediction platform, based on a model created based on a training set of molecules (hybrid method relying on fragments and topological descriptors) [100]. The pharmacokinetic properties are limited here to cytochromes (CYP) involved in the metabolism of xenobiotics because when considering the synergism of drugs and flavonoids, this group of proteins is the most important among the molecular targets included in the base sets of ADME prediction platforms [100]. Electrochemical parameters were calculated by optimizing the molecules in the Gaussian 09 program to have comparable data for all ligands. Values for DOX were obtained from literature data where the same computational chemistry method was used [86,101]. Based on the analysis of the results (Table 1), it can be seen that myricetin is not characterized by parameters that could determine the ability to inhibit the most important CYP isoenzymes. Other analyzed flavonoids have the ability to inhibit only selected isoenzymes, and chrysin can additionally inhibit CYP2C9, which is characterized by a relatively large volume of the active site pocket/cavity [102]. In the case of the reactivity of flavonoids, it can be seen that the smallest difference between HOMO and LUMO values is shown by myricetin and quercetin. This indicates higher chemical reactivity and low kinetic stability. Chrysin and apigenin are stable, and therefore they are chemically harder than other compounds with a greater energy difference HOMO-LUMO [103]. 

## 4. Anthracycline and Flavonoids Crosstalk

A unique feature of flavonoids as a group of chemical structures is their dualistic nature—antioxidant and prooxidant. Prooxidative properties mean the ability to intensively generate reactive oxygen species (ROS) through a number of different mechanisms, including inhibition of the complex I and II of the mitochondrial respiratory chain [104], as well as transformation into an unstable semiquinone radical undergoing oxidation to a reactive product, causing oxidative damage of DNA and lipid membranes [105,106].

Selected flavonoids with prooxidative potential may therefore exhibit related cytotoxicity and antiproliferative potential. Therefore, considering the structure and the pro- and antioxidant activity of chemical compounds determined by it is particularly important when designing a complex anti-cancer therapy. The interaction between chemotherapeutics and antioxidants is complex and involves a number of other mechanisms in addition to promoting and suppressing oxidative stress, and the final effect will depend on the dose used [107]. In the case of pro-oxidative activity, it is believed that it can be initiated by the coupling of transition metals present in body solutions, such as divalent copper and iron [108]. An interesting phenomenon is therefore the mechanism of DNA damage by polyphenols in the presence of transition metal ions, which is compared to the mechanism of action of conventional anticancer drugs, e.g., doxorubicin or cisplatin [105,109]. This unique property of flavonoids is the basis for exploiting the redox state of cancer cells, which are often characterized by a higher concentration of metal ions, in order to design stronger therapeutic strategies [110].

Considering both the structure and mechanism of action of flavonoids and anthracycline drugs, significant similarities in the molecular targets they act on, as well as participation in the same signaling pathways involved mainly in maintaining free-radical homeostasis, were noted [111,112]. 

It is known that the toxicity of anthracycline antibiotics is largely determined by the generation of intracellular oxidative stress [113]. The dysregulated redox balance of cancer cells resulting from the increased demand for ATP results in a more effective system of cell death avoidance regardless of the constantly higher levels of ROS compared to healthy cells. Cancer cell survival is also favored by higher intracellular levels of reduced glutathione (GSH) [114].

Consideration of the electron state of conventional drugs, including from the group of anthracycline antibiotics is also important in the process of discovering new derivatives of these drugs and predicting their molecular activity [86]. Due to the analyses of the stable conformation of DOX, it was determined which atoms (corresponding to HOMO orbitals) could spontaneously donate electrons and which would be responsible for their reception (LUMO) (Table 1). Comparing the values of physicochemical descriptors, it can be observed that DOX, compared to flavonoids, is characterized by high reactivity and low kinetic stability (lowest HOMO-LUMO energy gap). In addition, a high value of the dipole moment will indicate a higher solubility of DOX in polar solvents compared to flavonoids, and a higher value of electron affinity will also determine higher reactivity. Therefore, it seems important to consider certain relationships between the physicochemical properties of flavonoids and the synergistic effect of a drug with defined biological properties.

When describing the cross-talk of flavonoids and anthracyclines, the pathways of transport and metabolism of these molecules should also be considered. Pharmacokinetic drug interactions with flavonoids can modulate both P-gp transporters and CYP3A enzymes [115]. Both of these proteins are regulated by nuclear receptors, such as the pregnane X receptor (PXR)—a key regulator of xenobiotic detoxification [28], whose activity may be induced by redox-active compounds, including flavonoids [29].

### 4.1. Glutathione Depletion and Oxidative Stress

The concept of using antioxidants in anti-cancer therapy has been controversial for many years, and the observed effects are debatable. Epidemiological studies and many experimental results indicate a selective anti-cancer activity of popular antioxidant compounds, e.g., vitamin C, vitamin E and polyphenols, including flavonoids [116,117]. However, these conclusions cannot be extrapolated to all antioxidants. It has been observed that antioxidants such as *N*-acetylcysteine (NAC), which is widely used as the standard antioxidant in cancer biology, may play an oncogenic role [118]. Current research indicates, inter alia, that supplementation with selected antioxidant compounds may contribute to the initiation and growth of the tumor [119]. Glutathione (GSH)—the main buffer and the dominant cell thiol—is mainly responsible for maintaining redox homeostasis in the cell. The levels of this enzyme are elevated in many different cancer cells: breast, ovary and lung, and are associated with greater resistance to chemotherapy [120,121,122]. Interestingly, the reduction of intracellular GSH reduces the resistance of cancer cells to many cytotoxic drugs, which makes them more sensitive to oxidative damage [123].

It has been observed that phenolic compounds have the potential to deplete GSH in cancer cells and this effect is stronger in the case of flavonoids than hydroxychalcones (with are α,β-unsaturated ketone). The reaction of cells to phenolic compounds was different depending on the cell line. In the lung tumor cell line A549, the level of GSH after 24 h incubation with compounds at a concentration of 25 µM was chrysin < 5%, apigenin 9%, kaempferol 7% and quercetin 36% of the initial concentration. In the HL-60 and PC-3 lines, intracellular GSH concentrations after incubation with flavonoids were higher than in the A549 line, however, an increase was observed, respectively: chrysin > apigenin > kaempferol > quercetin [124]. Differences between cell lines may result from the existence of at least two different transport systems responsible for the preferential recognition of different groups of compounds, especially since the expression of MRP1 in cells was much higher in the A549 line than in the HL-60 or PC-3 line [124]. This may therefore indicate a relationship between the phenomenon of GSH depletion and the efficiency of transport of molecules by various MRPs [124].

Promising results were obtained in studies where the use of apigenin and quercetin together with DOX in leukemia cells led to a decrease in ATP levels and a decrease in GSH levels with an increase in markers of DNA damage [125]. Interestingly, other natural compounds, emodine or rhein, have an antagonistic or competitive effect on ATP levels and apoptosis, which was usually associated with an increase in GSH levels and no DNA damage [125].

In cellular (MCF-7, HepG2, Caco-2) and xenografted nude mice models, it has been observed that co-administration of an exogenous dose of glutathione (GSH) and DOX reduces the antitumor efficacy, as ROS production is one of the major mechanisms of anthracycline toxicity [126]. Recently designed by Xiao et al., polyprodrug conjugated with DOX leads to the effective depletion of GSH by disrupting redox homeostasis [127]. As a consequence, the IC50 value of the drug complex constructed in this way is three-fold lower against resistance to the MCF-7/ADR cell line compared to free DOX [127]. The action of quercetin will be strongly dependent on the availability of intracellular reduced GSH [128]. Under oxidative stress induced by drugs such as anthracyclines, in the presence of peroxidases, quercetin will react with hydrogen peroxide to form a semiquinone radical. The semiquinone radical is rapidly oxidized to quercetin-quinone, which is pro-oxidative and highly reactive toward protein thiols. It will react preferentially with GSH to form stable oxidized proteins (Figure 3). In low concentrations of quercetin, as a result of increasing their antioxidant capacity, proliferation may be induced, and at high concentrations or long exposure, a decrease in the antioxidant capacity and thiol content leads to cell apoptosis [128,129,130].

DOX is converted to its semiquinone radical form, leading to oxidative stress by transferring one electron to molecular oxygen (O^•−^). In this reaction, the electron donors are nicotinamide adenine dinucleotide phosphate (NADPH) molecules. Prolonged exposure to quercetin or its action at higher concentrations leads to the formation of quercetin-semiquinones and quercetin-quinones, with pro-oxidative and reactive properties. These products react easily with thiols and react with reduced glutathione (GSH), causing GSH depletion. The lack of GSH availability results in an ineffective hydrogen peroxide (H_2_O_2_) reduction reaction by antioxidant defense enzymes—glutathione peroxidase (GPx) and superoxide dismutase (SOD2). The disruption of GSH antioxidant defense in cells with persistent ROS overload, like malignant cells, leads to cell death by apoptosis.

Sensitization of cells to DOX by GSH depletion has also been observed for chrysin. The simultaneous synergistic effect of DOX and chrysin lowered the IC50 value of DOX to 43% in A549 cells, 47% in H157 and H1975 cells, and 78% in H460 cells [131]. The available evidence may therefore suggest that the effect of GSH depletion is determined by the structure of the flavonoid and depends on the number of hydroxyl groups, however, significant differences between the cell lines suggest the involvement of an additional factor, namely MDR1 membrane transporters, whose transport activity and the ability to hydrolyze ATP may be modulated by the presence of flavonoids [132,133].

### 4.2. Flavonoids As Inhibitors on Nrf-2/ARE Pathway

Thioredoxin (Trx) under the control of the transcription factor Nrf2 (nuclear erythroid 2-related factor 2) plays an important role in the regulation of redox homeostasis and chemoresistance by regulating the proliferation potential of cells [128,134] This system regulates the expression of genes (AP1—activator protein 1, p53—tumor suppressor p53, and NF-κB—nuclear factor kappa-light-chain-enhancer of activated B cells), inhibits the activity of the pro-apoptotic apoptosis signal-regulating kinase 1 (ASK-1) and participates in the detoxification of intracellular peroxides. Therefore, inhibitors of Trx and the Nrf2/ARE (antioxidant response element) pathway constitute an interesting target for the design of anti-cancer drugs [80]. Nrf-2 signaling has been observed to be associated with drug resistance, and cells exposed to DOX stimulate Nrf-2 signaling avoiding cell death [135]. Activating Nrf-2 signaling is beneficial in reducing the side effects of chemotherapy, but at the same time, this factor may mediate drug resistance, e.g., by strengthening the antioxidant defense system or by activating several oncogenes not directly related to antioxidant activity, e.g., metalloproteinase 9 (MMP-9); B-cell lymphoma 2 (BCL-2); B-cell lymphoma-extra large (BCL-x), tumor necrosis factor α (TNF-α); vascular endothelial growth factor A (VEGF-A) [136]. Many flavonoids like chrysin [137], apigenin [138] or quercetin [139] have been characterized as Nrf-2/ARE modulators, which is a promising area for research into synergistic compounds in anthracycline therapy [140].

It has been observed that chrysin quenches the ERK and PI3K/Akt pathway, which inhibits the expression of Nrf2 and elements of secondary signaling pathways (MRP5, multidrug resistance protein 5; HO-1), thereby increasing the sensitivity to DOX (Table 2) [137]. Studies have shown that in breast cancer cells (MCF-7), chrysin administered in the form of loaded nanostructured lipid carriers (NLCs) with higher cell uptake capacity induced apoptosis by inhibiting the Nrf-2 pathway, and also increased the sensitivity of these cells to DOX while increasing percentage of cells in sub-G1 phase [141] (Table 2). Interestingly, in glioblastoma cells, chrysin suppressed the expression of the Nrf2 protein and its major target genes as well as disrupting the ERK/Nrf2 signaling pathway without inducing kelch-like ECH-associated protein 1 (Keap-1) protein levels, the Nrf-2 negative modulator. In this case, the induction of apoptosis was carried out by stimulation of MAPK, regulating the activity of Nrf2 in a manner independent of Keap1 [142].

The inhibitory effect on Nrf2 apigenin expression in DOX-resistant hepatocellular carcinoma cells is due to enhanced expression of miRNA-101 reducing Nrf2 expression. This is a newly identified pathway involved in the proapoptotic action of apigenin, suggesting that other flavonoids may also regulate signaling pathways by modulating mi-RNA expression [143]. It was also observed that kaempferol at a concentration of 25 μM selectively lowered mRNA and the level of Nrf2 protein in A549 and NCIH460 cells and its target genes—NAD(P)H Quinone Dehydrogenase 1 (NQO1), heme oxygenase-1 (HO1), aldo-keto reductase family 1 member C1 (AKR1C1) and GST—but without effect on Keap-1. At the same time, long-term incubation with this flavonoid resulted in the accumulation of ROS, which was probably directly responsible for apoptosis [144].

Similar conclusions were reached by Wang et al., where in PANC-1 and MIA PaCa-2 cells, under the influence of kaempferol, there was an increase in ROS production, which resulted in a disruption of Akt/mTOR signaling with a simultaneous decrease in Keap1 levels [145]. The effect on Nrf2 was inconclusive—to some extent, an increase in its concentration was observed, but the main target of this compound turned out to be transglutaminase (TGM2), and its expression shows a negative correlation with ROS-dependent apoptosis. However, the opposite effect was observed in the HepG2-C8 cell line. A detailed comparison of the analyzed flavonoids is presented in Table 2, where the induced effects on the Nrf2/ARE signaling pathway in vitro and the observed effects in the downstream gene expression and modulation of Nrf2-dependent signaling pathways are listed.

Quercetin, kaempferol and pterostilbene activated the Nrf2/ARE signaling pathway and the effect of activating this pathway is synergistic when these compounds act simultaneously [147]. At the same time, the observed increase in NQO1 and SOD1 proteins is explained by the protective effect of these compounds against oxidative stress [147]. This protective effect was also observed when quercetin was orally administered to DOX-treated mice. As a result, concentrations of 10 mg/kg, 25 mg/kg and 50 mg/kg reduced histological abnormalities and reduced cardiomyopathy by increasing Nrf2 expression and increasing antioxidant defense [148]. In studies conducted in acute myeloid leukemia (AML) models, it was observed that treatment with quercetin for 48 h at a concentration of 50 µM, decreased the levels of histone deacetylase 4 (HDAC4), Nrf2 and phosphoro-Nrf-2 (p-Nrf2). At the same time, there was a decrease in the nuclear localization of Nrf2 and a decrease in HDAC4, a redox-sensitive deacetylase, which led to increased cell apoptosis [146]. 

So far published data on the effect of myricetin on the Nrf-2/are signaling pathway are scarce, but it is known that it regulates Nrf-2/HO-1 signaling in chondrocyte cultures in in vitro conditions [149]. In tumor cells, myricetin has been shown to be proapoptotic by downregulation of c-Myc in ovarian tumor cells [150], and, as is known, c-Myc directs malignant tumor progression by Nrf-2 [151]. This may suggest some indirect influence on the regulation of redox-dependent factors by myricetin in neoplastic cells. Qin et al. indicate that in HepG2 carcass myricetin activates the Nrf2/Are pathway by inhibiting ubiquitination of this factor, which leads to an increase in nuclear Nrf2 accumulation and enhancement of ARE-dependent gene expression [152]. 

### 4.3. Flavonoids As Agonist and Antagonist of Nuclear Receptors

The observed effect of flavonoids on drug activity may include certain interactions taking place in the cell nucleus, namely antagonistic and agonistic activity toward nuclear receptors crucial in the regulation of xenobiotic metabolism. In drug–drug and drug–flavonoid interactions, pregnane receptors X receptor (PXR), constitutive androstane receptor (CAR), and aryl hydrocarbon receptor (AhR) seem to be particularly important, as they are mediators induced in exposure to exposure to conventional drugs, bioactive plant metabolites [153] or environmental pollutants [154]. They act as modulators of gene transcription of cytochrome P450 isoenzymes. Together with the genes encoding the CYP isoenzymes, these receptors form a related network of linkages in which the enzymes responsible for the metabolism of xenobiotics regulate the response leading to the induction of the expression of the genes encoding these enzymes [155].

AhR is a member of nuclear hormone receptors, which is activated by polycyclic organic compounds, but more than polyphenols, plant alkaloids [156] and selected drugs [155]. PXR belongs to the group of orphan receptors, which participates in the induction of, among others, CYP3A, CYP3B and selected transporters from the MDR proteins. PXR and CAR are called “xenobiotic sensors” and are able to bind to structurally different ligands, regulating the expression of genes encoding phase I and II enzymes and transporters involved in the detoxification and elimination of xenobiotics [151]. Thus, the pharmacological effect of anthracycline antibiotics depends indirectly on the activity of nuclear receptors that regulate the expression of membrane transporters. It has been observed that among patients with breast cancer, the clearance of doxorubicin is significantly lower in the case of the PXR gene haplotype resulting in lower expression of the enzyme itself and its downstream targets involved in drug metabolism, CYP3A4 and ABCB1 [157]. Therefore, it can be expected that the inhibition of the activity of nuclear receptors will result in a decrease in the clearance of the drug and its higher concentration in the plasma, however, in the case of flavonoids, this effect is not clearly defined. Bioflavonoids have so far been extensively studied as AhR, PXR or CAR ligands, which exhibit both agonist and antagonist activity, which depends on the structure of the compound itself and the cell model used in the research [158].

In the case of chrysin, it has been observed to act as an indirect CAR activator to enhance nuclear translocation in primary human hepatocytes [159]. Apigenin has also been reported as an indirect activator of the PDR via cyclin-dependent kinase 5 (Cdk5) in human HepG2 liver cancer cells, however, the ability of the flavonoid to interact directly with the PXR has been ruled out [160].

Interestingly, the most prominent structural feature described by Jin et al. influencing AhR agonist or antagonist activity was the number of hydroxyl moieties that form hydrogen bonds with AhR residues [156]. However, the position of the hydroxyl groups and complex specific interactions may play an equally important role. In the studies, Kaempferol showed antagonistic activity leading to a decrease in the activity of CYP1A1 and CYP1B1, while the action of quercetin in epithelial cells MCF-10A resulted in the activation of AhR and an increase in the expression of both of these transporters [161,162]. Very large discrepancies between individual experimental models make it impossible to put forward a hypothesis about the relationship between the structure of the analyzed flavonoids and the interaction with AhR, PXR and CAR receptors, however, the confirmed effect of this group of compounds on the activity of this group of proteins emphasizes the importance of direct and indirect regulation of drug metabolism pathways and xenobiotics [163,164,165].

## 5. Modulatory Effect of Flavonoids on the Activity of Multidrug Response Proteins

One of the most frequently proposed approaches to synergism and counteracting drug resistance is targeting the inhibition of efflux proteins as well as cytochromes involved in the metabolism of xenobiotics [166]. DOX is metabolized to the major metabolite doxorubicinol by P-gp, CYP2D6 and CYP3A4. The CYP3A4 isoform is largely responsible for the elimination of phytochemicals in food and the majority of medicines. Food compounds, among which the group of polyphenols has been most thoroughly studied, show the ability to interact with CYP3A4 and alter its expression and activity [167,168].

The family of OATP transporters is involved in the uptake and distribution of structurally divergent endogenous and exogenous substrates, including steroid hormones, bile acids and drugs used in anti-infectious diseases, antihypertensive diseases and in recent years, including anti-cancer drugs [169]. OATP1A/1B transporters are expressed in many neoplastic tissues—breast, colon, ovary, lung, prostate and bone cancer [170]. OATPs are also a class of transporters that will determine the efficiency of flavonoids transport, and therefore the polymorphism of genes encoding this group of proteins will determine, at the same time, the change in the pharmacokinetic properties of drugs and other xenobiotics and the interaction between them [171,172].

### 5.1. Modulation Effect of Flavonoids on Cytochrome P450 3A4 (CYP3A4) Activity

Interactions between polyphenols and CYP3A4 have proven clinical consequences, e.g., in the form of potentiation of the effect, including side effects of repeatedly used drugs [167]. Unfortunately, we still have limited knowledge of the transporters that interact with flavonoids and the structural features of the flavonoids that would make them act as substrates or modulators of a given transporter [173]. Understanding the flavonoid transport route is a topic that requires very careful study when considering them as potential enhancers for conventional drugs. Both drugs and flavonoids may use the same transport route or, on the contrary, may not inhibit their transport [174].

It has been observed that the use of a xanthan compound with known properties to inhibit the CYP3A family by inhibiting the metabolism of DOX may increase its plasma concentration. Miladiah et al. suggested that this may explain the observed synergistic effect [175]. Interestingly, under the hypoxia conditions in HepG2 cells, a decrease in DOX cytotoxicity is observed, which is correlated with an increase in CYP3A4 expression [176]. However, we currently have too little data to talk about a phenomenon that can occur in all types of cancer cells. Higher CYP3A4 expression is generally noted in cancer cells compared to normal tissues as well as in primary tumors of distant metastatic patients compared to non-metastatic patients. Perhaps the overexpression of CYP3A is responsible for the inactivation of cytotoxic substances, allowing the rapid proliferation of cancer cells [177,178,179].

Current data indicate that the action of natural compounds from the flavonoid group may modulate the activity of CYP—both activate and inhibit and thus regulate the bioavailability of the drug. Basheer et al. suggested that flavonoids could be considered as P450 inhibitors by direct covalent bond formation leading to their inactivation or reversible bond formation as well as changes in the expression of P450 enzymes [167]. There is conflicting evidence for the modulation of CYPs by flavonoids observed in in vitro and in vivo conditions, however, the reason for these differences is not clearly explained. Quercetin has been observed to stimulate CYP3A4 mRNA expression in the human colorectal adenocarcinoma cell line (Caco-2) and in primary human hepatocyte lines [180,181]. An increase in the bioavailability of orally administered DOX was observed with the concomitant action of quercetin (15 mg/kg, oral dose, rats) by inhibiting P-gp and limiting first-pass metabolism by inhibiting CYP3A in the small intestine/liver [182]. In the case of intravenous administration of DOX to rats, quercetin did not significantly affect the pharmacokinetics of the drug, which is explained by the metabolic instability, rapid biotransformation of quercetin and achieving very low concentrations in vitro [183]. However, in the case of the immunosuppressant—cyclosporin administered orally, it has been observed that quercetin reduces the bioavailability of the drug by activation of P-gp and CYP3A (50 mg/kg of quercetin, oral dose, rats) [184]. The opposite effects of quercetin may result from the activation of the enzyme not directly by quercetin, but by the product of its metabolism—in the sulfated or glucuronidated form [167].

In the case of myricetin, the effects on CYP3A4 activity are also not the same in the research models used. In rats undergoing oral DOX therapy, an increase in bioavailability of the drug was observed in the presence of this flavonoid (0.4, 2 and 10 mg/kg), which is due to the inhibition of P-gp and the reduction of metabolism by CYP3A. A similar relationship was not observed when the drug was administered intravenously [185] (Table 3). This observation highlights the importance of the route of administration, i.e., a different path of flavonoid metabolism in shaping the final effect on the activity of the CYP3A4 cytochrome. In addition, the observed different effects of flavonoids on DOX in terms of CYP3A4-regulated metabolism may result not only from direct reactions with the enzyme but also from differences in the expression of CYP3A4 in different cells and tissues, which is regulated, among others, by exposure to xenobiotics [181].

Unfortunately, the available data describing the activity of all analyzed ligands against CYP3A4 are insufficient to determine the pharmacophore responsible for the inhibitory effect of this class of compounds on CYP3A4 [167]. However, by using simple in vitro models using drug-resistant human cell lines overexpressing P-gp [182,185,186,187,188] as well as insect cells with catalytically similar to human liver microsomal CYP3A4 [189] and monitoring the marker reaction of CYP3A4 enzyme with flavonoids [187], it is possible to note some simple structure–activity relationships (Table 3). On the basis of the structure analyses, it was observed that for the inhibitory activity on CYP3A4 of the analyzed compounds it was necessary to substitute in group 5 in ring A (present in all analyzed compounds) and monosubstitution of ring B (in apigenin) (Table 3) [187]. This group does not undergo ionization, and the formed hydrophobic interactions with the enzyme are probably crucial for the inhibitory activity [182]. In the case of disubstituted OH groups in the 3′ and 4′ positions, no effect on the enzyme activity was shown [187]. Another important feature of the structure ensuring stronger binding of the substrate is high lipophilicity, the pattern of which is in the order myricetin → chrysin (Table 1) [190]. 

### 5.2. Flavonoids Interactions with Organic Anion-Transporting Polypeptide (OATP)

The OATP1B1 polypeptide is a liver-specific uptake transporter important in the distribution of drugs in the liver, and inhibition of hepatic drug uptake via OATP1B1 by other drugs or flavonoids may result in clinically significant interactions between these compounds [191]. Experimental results show that OATP1B transporters play a significant role in hepatic uptake, clearance and plasma exposure of DOX and facilitate drug transport [170]. In vitro, in vivo and in silico studies show that the critical pharmacophores in the flavonoid structure for inhibiting OATP1B1 are hydrogen bond acceptors and hydrogen bond donors at 4′, 5 and 7 positions [64]. The most potent inhibitor of this transporter in HEK293 cells turned out to be licochalcone (IC50 = 7.96 μM), a natural phenolic compound belonging to chalconoids, and among flavonoids, luteolin without a hydroxyl group in the B ring, it turned out to be stronger (IC50 = 22.03 μM) than kaempferol (IC50 = 33 μM) [192]. Due to in silico models, it has been observed that aglycones of flavonoids will interact more strongly with OATP1B1 than glycosides, and the presence of one or two hydroxyl groups in ring B (apigenin → quercetin; Figure 1) will be beneficial for hydrogen bonding with this protein [191]. In vitro studies with the use of the competitive substrate BSP sulfobromophthalein (BSP) showed that the IC50 inhibition value on OATP1B1 protein by quercetin, kaempferol, and apigenin were 8.7, 15.1, 20.8 µM, respectively [193].

In the case of an isoform OATP1A2, polymorphs of the gene of this transporter may influence the observed inter-individual variability in response to this anthracycline [170]. DOX as a weak hydrophobic base is not a typical OATP substrate, however, cells overexpressing OATP1A2 show about two-fold higher drug uptake [136]. It was observed that inhibition of the OATP1A2 transporter by naringin, a component of grapefruit juice, resulted in a significant reduction in the cytotoxicity of breast cancer cells treated with DOX [136]. In the case of inhibition of the OATP1A2 protein, the IC50 values obtained in in vitro experiments were higher than for OATP1B1, but the inhibitory potential decreased in a series of quercetin > kaempferol > apigenin and IC50 values were 22, 25.2 and 32.4 µM, respectively [193].

Observing unambiguous relationships is not easy in biological, complex models or in vivo conditions, because the activity of antagonistic activity will also be influenced by external, physicochemical factors such as environmental pH. For example—in an acidic reaction, quercetin will be a substrate for OATPB, while at alkaline pH, it will be transported by passive diffusion [193]. Then, the lack of interaction of quercetin with OATP is not due to the flavonoid structure per se, but to a solvent effect. The interaction of flavonoids with OATP transporters and the participation of these transporters in the distribution of anthracyclines is a topic that requires careful analysis. This applies in particular to the design of potential enhancers based on the flavonoid structure that may exert, inter alia, a beneficial hepatoprotective effect by inhibiting OATP, but at the same time limiting the effectiveness of the chemotherapeutic agent.

### 5.3. Role of Flavonoids in ABC Transporters Activity in Cancer Cells

ATP Binding Cassette (ABC) Transporters are one of the largest families of transmembrane proteins found in all living organisms. Most of them act as active transporters responsible for the transport of substrates across the cell membrane against the gradient of concentrations using energy from ATP hydrolysis [194]. Transporters of this subfamily of proteins export compounds with a very diverse chemical structure, reducing the intracellular concentration of the drug [195]. Of the 48 proteins of ATP-binding cassette protein family encodes in the human genome, three closely related proteins: P-gp (ABCB1), MRP1 (ABCC1) and BCRP (ABCG2) are known to be responsible for resistance to anthracycline drugs [196].

Overexpression of ABC transporters in cancer cells is one of the most important mechanisms of multi-drug resistance (MDR) [194]. These transporters actively translocate drugs reaching the cell through the cell membrane, which reduces their intracellular accumulation and significantly reduces the therapeutic effect [197]. For this reason, the concept of using ABC efflux pump inhibitors together with the administered chemotherapeutic agent is a promising strategy for restoring the drug sensitivity of resistant tumor cells or improving the pharmacokinetic profile [198].

Potential inhibitors of efflux mediated by ABC class transport proteins include polyphenols of plant origin, including flavonoids such as chrysin [199], apigenin [200], kaempferol [201], quercetin [202] or myricetin [203]. Some researchers also undertook the subject of chemical modification of these flavonoid ligands in order to strengthen the inhibitory potential against ABC class transport proteins. Among these modifications is dimer apigenin with five or six ethylene glycol units with 6-methyl or 7-methylsubstitution on the A ring [204]. It has been observed that the synthesized apigenin homodimer effectively inhibits DOX efflux and thus reverses the MDR by high affinity/competitive inhibition of binding within the MRP1-DOX substrate binding site [204]. Through the use of high-throughput screening, flavonoid dimers were defined that exhibited high selectivity toward MRP1, as well as inhibitory potency up to 36-fold greater than the known MRP1 inhibitor verapamil [204]. 

Apigenin itself in the form of an aglycone also showed a visible effect on drug-resistant cells in vitro. Interestingly, it has been observed that drug-resistant cells with overexpression of ABC transporters: BCRP/ABCG2, P-gp/ABCB1 show increased cellular uptake of DOX after apigenin treatment [200]. More detailed studies including docking studies indicate that apigenin inhibits DOX efflux transport l by competitive inhibition of nucleotide-binding domains which is responsible for ATP hydrolysis. This results in the cleavage of ATP leading to energy depletion and increased drug accumulation inside the cell [200].

Quercetin inhibits both ABCB1 gene expression and P-gp function in the RDB pancreatic line, acting as a sensitizer of tumor cells to daunorubicin. After 72 h of treating RDB cells with quercetin at a concentration of 12 µM, a 30% decrease in P-gp was observed [205]. The effect of this flavonoid is also evident in in vivo tests. Quercetin taken at a dose of 500 mg per day has been shown to significantly induce P-gp/MDR1 activity, although the authors emphasize that both quercetin and its main metabolites can to some extent induce P-gp activity, affecting the final pharmacological effect of the taken drugs [206]. Using computational tools and molecular dynamics (MD) simulations, it has been shown that the effects of quercetin can result from binding to the interface of the ICH2 (intracellular helices) and NBD2 (nucleotide-binding domains) region of P-gp and thereby modulates its functional activity [207].

The BCRP protein, also called ABCG2, is another representative of the ABC transporters, which plays an important role in tissue protection through the active transport of endogenous substrates as well as xenobiotics [208].

BCRP is a protein involved in the transport of physiological substances such as estrone-3-sulfate, 17β-estradiol 17-(β-d-glucuronide) and uric acid, as well as chemotherapeutic drugs [209]. In normal tissues, this transporter is involved in the distribution of the elimination of xenobiotic substances, protecting cells from exposure to carcinogens [210]. However, BCRP, as a protein involved in the active outflow of anti-cancer drugs, contributes to the occurrence of MDR; therefore, it is expected that blocking BCRP activity may provide better treatment outcomes. The pharmacophore model indicated that the aromatic ring B, hydrophobic groups and hydrogen bond acceptors may play a key role in the inhibitory power of BCRP by flavonoids [211].

Flavonoids as agonists of the AHR receptor (aryl hydrocarbon receptor), which is a transcriptional activator of BCRP, can induce the expression of this transporter in Caco-2 cells, which indicates that they support the detoxification of food-derived procarcinogens by compounds such as quercetin, chrysin, flavone, and indole-3-carbinol [100]. 

The effectiveness of the efflux of flavonoids as xenobiotics through the BCRP transporter outside the cell depends on the chemical form of the metabolite in which the flavonoid occurs in the body. In an experiment using selective chemical inhibitor of BCRP, Ko-143, the chrysin 7-O-sulphate efflux was inhibited, but without affecting chrysin glucuronide efflux [196].

When assessing the impact of flavonoids on the transport efficiency mediated by BCRP, one should also consider the indirect impact on the activity of this class of proteins, whose expression depends on many factors, e.g., conditions of hypoxia and normoxia. The transcription of the BCRP gene is regulated by many factors, including HIF1α (Hypoxia-inducible factor 1) [212], regulating the hypoxic responses. The efficacy of DOX has been shown to be significantly limited in the hypoxic microenvironment (pO_2_ ≤ 2.5 mmHg), which is usually found in the central region of solid tumors [213]. Consistent with the assumption that the hypoxic state increases the resistance of breast cancer cells to DOX and the hypoxic state itself induces HIF-1α overexpression, targeting HIF-1α may overcome therapeutic resistance. It has been proven that, among others, chrysin contributes to the stimulation of HIF 1α degradation by ubiquitin-proteasome and modulates its expression through the PI3K-Akt pathway in neoplastic cells [214], thus it may indirectly affect the activity of HIF1 α -dependent transport proteins. In the case of the simultaneous action of chrysin as a BCRP/ABCG2 inhibitor, an increase in the toxicity of the cytostatic drug sorafenib was observed in the HCC cell line (Hep3B and HepG2) [215]. In Caco-2 cells, the enhancement of BCRP expression was demonstrated by chrysin by enhancing binding to the AHR as an actuator of BCRP protein transcription [216].

## 6. Genotoxicity vs. Genoprotection in Anthracycline-Treated Cells

There are a lot of data indicating the protective effect of flavonoids on cells exposed to the toxic effects of anthracycline drugs [217,218,219]. In the case of non-malignant cells, this effect is highly desirable, taking into account, inter alia, the high cardiotoxicity of DOX by damaging the functionality of the cardiomyocytes [220]. The cardiotoxicity of this drug may result from the direct effects of pro-oxidative reactions, especially after iron binding, and by initiating mitochondrial disturbances [221]. The protective effect of phenolic compounds on healthy cells may enable the use of a chemotherapeutic drug in effective concentrations. What does it depend on, then, whether a given compound can act as a protective compound against DNA damage or can cause this damage itself? The dualistic nature of the activity of phenolic compounds leads to a comprehensive consideration of this issue depending on the conditions under which a given effect is observed [222].

Chrysin has been observed to counteract the effects of DOX-induced oxidative stress, in the rat testicle genotoxic damage model, by reducing defective sperm count [223]. On the other hand, the chronic effect of chrysin on DNA damage may indicate a potential drug-reducing effect, as one of the mechanisms of action of DOX is the formation of DNA adducts [224]. Earlier studies showed that flavonoids such as chrysin, apigenin and quercetin do indeed have the ability to protect against DNA damage, including inhibiting benzopyrene and IQ (2-amino-3-methylimidazo [4,5-f] quinoline) induced adduct formation [225]. 

Comparing the affinity of the ligand to the DNA double helix in solutions, it was found that the flavonoids themselves have the ability to bind to DNA, with apigenin having a higher affinity than polyphenols having an additional hydroxyl group in the B ring (morin 3,5,7,2′,4′-pentahydroxyflavone) and naringenin glycoside (naringin) (4′,5,7-)trihydroxyflavone-7-rhamnoglucoside). DNA binding constants (K), which were determined directly from the concentrations of the free ligand, DNA and DNA in the form of a ligand-DNA complex, were estimated as, K_apigenin_ = 7.10 × 10^4^ M^−1^ > K_morin_ = 5.99 × 10^3^ M^−1^ > K_naringin_ = 3.10 × 10^3^ M^−1^ [225]. Although earlier studies by Zhang et al. indicate the possibility of intercalation of apigenin to DNA, the values of Tm, ΔTm (melting temperature) obtained in later Waihenya et al. studies exclude that this flavonoid may intercalate into dsDNA [226]. 

Through the use of research methods in cell-free models, a possible binding mechanism was identified, and it was suggested that chrysin may bind to DNA by partial intercalation without electrostatic interactions [227]. Molecular docking studies have shown that the lowest energy, i.e., the most advantageous type of apigenin binding occurs with a smaller DNA groove, and the formed complex is stabilized by several hydrogen bonds [226]. Due to the use of electrospray ionization mass spectrometry (ESI-MS), it was concluded that the 4’-OH group of flavonoids present in, among others, apigenin, quercetin, kaempferol, myricetin is important for binding to the DNA duplex [228] while comparing several flavonoids with each other, it can be observed that the presence of 3′ and 4′-OH, such as in quercetin and myricetin, will determine the lower binding energy [140]. 

Other studies showed that DNA damage in K562 cells in the comet test fell in line with myricetin > kaempferol > quercetin > luteolin > apigenin; however, the quercetin with the highest binding to DNA is not the most toxic [140]. Quercetin tends to form hydrogen bonds with thymine bases, although it is not excluded that apart from electrostatic interactions, there may be other binding mechanisms, e.g., various types of intercalation and bonding in the DNA grooves [229] (Table 4).

As Korga-Plewko suggests, both flavonoids and DOX undergo redox cycles, so there may be some kind of interaction between the radical forms formed [230]. Interestingly, the higher strength of quercetin binding to DNA occurs at acidic pH, i.e., the reaction corresponding to the tumor microenvironment. This is due to the fact that at a lower pH, the hydroxyl group on the B-ring remains in an undissociated state, so it will be more hydrogen-bonded with DNA, which provides a higher bond strength [140]. In addition, the presence of copper ions increases the cytotoxicity of quercetin by directly increasing the degree of DNA damage or inhibiting catalase, leading to pro-oxidative cell damage [140]. The research conducted by Mizutani et al. indicates that anthracyclines such as DOX, amrubicin, aclarubicin and pirarubicin induce oxidative DNA damage in the presence of copper(II), and as a result of the anthracycline oxidation reaction, a semiquinone radical, and the formed Cu(I)OOH and OH• radicals can cause DNA damage [231]. In the case of quercetin, it is also said to have a high pro-oxidative potential, which is probably due to the presence of a catechol moiety and the resulting susceptibility to auto-oxidation, leading to conversion to ortho-semiquinone and ortho-quinone, which react to Cu(I), forming a highly reactive OH• radical [232]. So, quercetin having a catechol group induces more oxidative damage than kaempferol, its analog (without catechol groups), in the presence of copper ions [233]. 

## 7. Conclusions

The group of bioactive flavonoids is currently considered not only as pro-health compounds of plant origin with chemopreventive effects but also as starting structures for the design of new drugs or combination therapy enhancers. The article focuses on five flavonoids that differ in the presence of one hydroxyl group. Based on the analyzed reports, it is indicated that the minimal structural difference entails diametrical differences in biological activity, although there is no data comparing the logical groups of phenolic compounds in the context of synergistic therapy with anthracyclines. Particularly noteworthy is the pleiotropic/multidirectional effect of flavonoids ranging from the direct antioxidant effect through the global modulation of signaling pathways. Despite the relatively low bioavailability, the action of flavonoids on membrane proteins is often sufficient to exert a synergistic effect with a chemotherapeutic drug.

It is generally believed that the concentration of a drug in the plasma is proportional to the therapeutic effect and toxicity of a given pharmaceutical, therefore some methods of increasing to a higher concentration in the body may be beneficial, taking into account the individual threshold of toxicity [234]. Of course, co-administration of a flavonoid or other bioactive plant metabolite carries potential risks, and more thorough pharmacokinetic and structure–activity relationship (QSAR, SAR) studies are needed to evaluate potential interactions with anthracyclines. To date, too few human case reports and clinical trials have been published reporting the effects of the concomitant use of individual flavonoids with anthracyclines. However, attempts to enhance the effects of therapy by including plant compositions have been made for a long time, the most popular of which is the Aidi injection (Z52020236, approved by China Food and Drug Administration), consisting of plant extracts of ginseng, *Astragalus membranaceus* and *Acanthopanax* [235]. In a 2020 meta-analysis by Xiao et al. of 80 randomized trials of Aidi injections with chemotherapy in the treatment of lung, liver, colorectal and gastric cancer, it was shown that in many cases, this combination significantly improved clinical efficacy and survival time of patients. In addition, an overall significant reduction in the incidence of hepatorenal toxicity side effects was observed [236]. Limited evidence and lack of studies on the effect of a single compound in combination with a drug significantly limit the potential clinical use of flavonoids in a preventive and therapeutic context.

Flavonoids are molecules with a general pleiotropic effect and their effect on the redox state of a cell can be described as dose-response hormesis, depending on the concentration and the cellular environment. At low concentrations, they act as antioxidants, activators of Nrf2/ARE signaling, and at higher concentrations, as prooxidants and inhibitors of this pathway [143,144]. The KEAP1-Nrf2 system is the main node that senses and reacts to redox disturbances resulting from internal and external factors [140]. The mechanism of the dual nature of Nrf2 is due to its ability to reduce carcinogen-induced electrophilic stress, thus DNA damage, and activating detoxification metabolism. Initially considered a tumor suppressor, it is now considered a proto-oncogene, as the constitutive regulation of Nrf-2 and its effectors is a signal stimulating survival in many aggressive types of cancer. Oxidative stress is induced, among others, by anthracillins or/and the presence of electrophilic compounds induces the activation of the Nrf2 factor, which binds to the ARE promoter region, leading to the transcription of antioxidant defense genes and phase 2 detoxifying enzymes. Due to the widely developed antioxidant defense mechanisms in cancer cells, targeting GSH depletion during therapy with anthracyclines also seems promising. Quercetin by reaction with free radicals is chemically converted to quinone, which easily reacts with GSH, thus forming two glutathione adducts at the C6 position and the C8 position of ring A.

Considering free radical processes, we also analyzed the phenomenon of GSH depletion, as a strong association between the high availability of this intracellular antioxidant enzyme [125] and carcinogenesis [119] was observed both in vitro [126] and in vivo [123] models. In the case of developing flavonoid derivatives with improved bioavailability, the strategy of GSH depletion during anthracycline therapy is a promising approach that can significantly reduce drug resistance and adverse effects caused by relatively high doses of the drug.

Another important mechanism that will significantly modulate the anticancer activity of anthracycline drugs is the regulation of membrane transporters belonging to ABC class proteins, which is a common reflux pathway for both anthracycline drugs and selected flavonoids [199,200,201,202,203]. It turns out that flavonoids can inhibit the activity of these proteins directly by allosteric inhibition, but also by indirectly depleting ATP and reducing the activity of proteins dependent on ATP hydrolysis [200].

Flavonoids in the form of aglycones as well as in the form of metal complexes can also bind to DNA, causing genotoxic effects comparable to the mechanisms of action of conventional chemotherapeutic drugs. Analyzing the transport routes and the biological effect caused, we observe the existence of certain dependencies among the analyzed ligands, although this effect is not always correlated with the number of hydroxyl groups or the antioxidant potential. The comparison of physicochemical descriptors gives a picture of differences in solubility, lipophilicity or electron charge distribution, which in biological systems will determine the strength of interaction and binding with macromolecules such as DNA, signaling proteins or cell membrane structures [222,226].

We believe that future research should therefore integrate computational chemistry, molecular modeling, genetics and molecular biology methods in a comprehensive way. This approach makes it possible to precisely define the relationship between the structure of phenolic compounds and their synergism with anthracycline drugs.

## Figures and Tables

**Figure 1 ijms-24-00391-f001:**
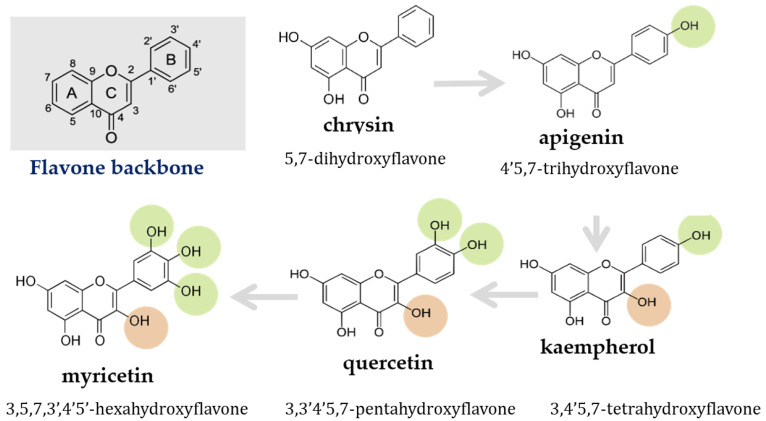
Structures of the analyzed series of flavonoids and the general formula of the flavone backbone with the designation of rings and numbering of carbon atoms. Hydroxyl substituents on ring B (green), substituent at C3 position on ring C (red).

**Figure 3 ijms-24-00391-f003:**
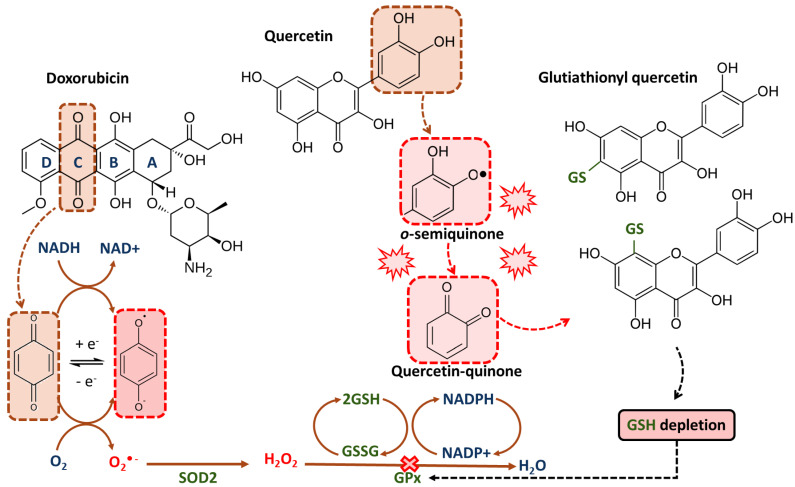
An increase in oxidative stress induced by inhibition of glutathione reductase may enhance the cancer response to chemotherapy by flavonoid-induced glutathione depletion.

**Table 1 ijms-24-00391-t001:** Physicochemical and structural properties of selected flavonoid.

		CHRY	API	KAE	QUE	MIR	DOX
Physicochemical Properties ^1^	Molecular Weight	254.24	270.24	286.24	302.24	318.24	543.52
	Volume	206.92	216.03	224.05	232.07	240.08	
	Aromatic heavy atoms	12	16	16	16	16	12
	H-bond acceptors	4	5	6	7	8	7
	H-bond donors	2	3	4	5	6	12
	TPSA	70.67	90.9	111.13	131.36	151.59	206.08
	SILICOS-IT Log P ^2^	3.02	2.52	2.03	1.54	1.06	1.17
	ESOL Log S	−4.19	−3.94	−3.31	−3.16	−3.01	132.66
	Molar Refractivity	71.97	73.99	76.01	78.03	80.06	131.52
Pharmacokinetics ^1^	CYP1A2	+	+	+	+	+	−
	CYP2C19	−	−	−	−	−	−
	CYP2C9	+	−	−	−	−	−
	CYP2D6	−	+	+	+	+	−
	CYP3A4	+	+	+	+	+	−
DFT electrochemical parameters ^3,4^	HOMO [eV]	−6.43	−6.24	−5.78	−5.65	−5.71	−5.32
	LUMO [eV]	−2.04	−1.88	−1.97	−1.97	−2.06	−4.09
	HOMO-LUMO energy gap [eV]	4.39	4.36	3.81	3.68	3.65	1.23
	Dipole moment	3.84	2.47	5.47	4.23	6.78	7.77
	Electron affinity	4.24	4.06	3.87	3.81	3.88	6.80

^1^ Obtained from SwissADME prediction [100] ^2^ Average value of predicted Log P (SILICOS-IT) model created based on training set of molecules (hybrid method relying on fragments and topological descriptors). ^3^ Obtained from Gaussian 09 computer software, based on optimized structures; own calculations (DFT, B3LYP base). ^4^ DOX: HOMO, LUMO, Electron affinity obtained from [86], dipole moment [101]. (+) is the predicted ability of the compound to inhibit the cytochrome isoform, (−) means no predicted CYP inhibiting property.

**Table 2 ijms-24-00391-t002:** Inhibition effect on Nrf-2/ARE pathway in cancer models.

Compound	Concentration	Cell Line	Cell Type	Effect	Ref
Chrysin	10–20 µM	BEL-7402/ADM cells	Hepatocellular carcinoma	↓ mRNA and protein expression of Nrf2, HO-1, MRP5, and aldo-keto reductase family 1 member B10 (AKR1B10)	[137]
Chrysin [nanostructural lipid carriers]	5–50 µM	MCF-7	Breast cancer	↓mRNA and protein expression of Nrf2; NQO1, HO1 i MRP1	[141]
Chrysin	10–60 µM	T98, U251, U87	human glioblastomas	↓ Nrf2, NQO-1, HO-1 (Keap1-independent); ↓ ERK signaling	[142]
Apigenin	10 μM	BEL-7402/ADM	Hepatocellular carcinoma	inhibiting miR-101/Nrf2 pathway	[143]
Kaempferol	25 μM	A549, NCIH460	NSCLC	↓ mRNA and protein expression of Nrf2; ↓ AKR1C1, NQO1, HO1 i GS; ↑ ROS	[144]
	25–50 μM	PANC-1, PaCa-2	pancreatic cancer	ROS-dependent suppression Akt/mTOR signaling; ↓ Keap; Nrf2 (ambiguous impact)	[145]
Quercetin	50 μM		human xenograft acute myeloid leukemia (AML) models, and in vitro using leukemia cell lines	↓ Nnf2 nuclear localization; ↓ pNrf2 ↓ HDAC4;	[146]

The arrow ↑ or ↓ indicates an increase or decrease in concentration/expression or content in the cells, respectively.

**Table 3 ijms-24-00391-t003:** IC50 values (μM) on inhibition of CYP3A4 cytochrome in cell in vitro model and cell-free assay.

		Cell Internalization of Rhodamine-123 in MCF-7/ADR ^1^ Cells Overexpressing P-gp	Ref	Human CYP3A4 Microsomes Expressed in Baculovirus ^2^-Insect Cell, Lineweaver–Burk Plot Analysis	Ref	Residual Activity Compared to Control (6β-Hydroxylation of Testosterone-Marker Reaction of CYP3A4 Activity)	Ref
Flavonols	Myricetin	7.8	[185]	44.5	[186]	133 ± 35	[187]
	Quercetin	1.97	[182]	22.1	[186]	126 ± 10	[187]
	Kaempferol	8.6	[182]	8.8	[186]	101 ± 14	[187]
Flavones	Apigenin	1.8	[182]	0.4	[186]	24 ± 3	[187]
	Chrysin	nd		0.9	[186]	17 ± 3	[187]

^1^ MCF-7/ADR—multidrug-resistant breast cancer cell model; ^2^ insect cells infected with a recombinant Baculovirus.

**Table 4 ijms-24-00391-t004:** Binding constants of flavonoids to DNA in cell-free assays.

	Binging Constants (K[M^−]^])	Methods	DNA Model	Measurement Conditions	References
CHRYSIN	1.21 × 10^5^	Abs	d(CCAATTGG)2	room temperature; absorption: 200 to 800 nm; fluorescence: Ex 326 nm, Em spectra 320 to 650 nm,	[227]
	9.07 × 10^3^	Fluo			
	9.03 × 10^5^	Abs	Ct-DNA		
	3.03 × 10^5^	Fluo			
APIGENIN	7.10 × 10^4^	FT-IR & Abs	Ct-DNA	FT-IR: sharp DNA band at 968 cm^−1^ as internal reference, obtained difference spectra [(DNA solution + ligand)-(DNA solution)]; Abs: of DNA at 260 nm, wavelength range of 245–445 upon titration of dsDNA	[225]
	9.12 × 10^4^	Abs	dsDNA		[226]
	0.73 × 10^4^	Abs	Ct-DNA		[140]
QUERCETIN	17.54 × 10^5^	Fluo, 300 K	Ct-DNA	fluorescence: Ex 220–660 nm 57; Em 240–680 nm	[229]
	8.97 × 10^5^	Fluo, 310 K			
	4.80 × 10^5^	Fluo, 320 K			
	18.0 × 10 ^4^	Abs	Ct-DNA		[140]
KAEMPFEROL	18.0 × 10^4^	Abs	Ct-DNA	measurement of absorption between 200 nm and 500 nm	[140]
MYRICETIN	8.63 × 10^4^	Abs	Ct-DNA		

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
