# Peer review of "Why Do Dietary Flavonoids Have a Promising Effect as Enhancers of Anthracyclines? Hydroxyl Substituents, Bioavailability and Biological Activity"

_ijms, 2022, doi:10.3390/ijms24010391_

Round 1

Reviewer 1 Report

However, the manuscript delivers a good topic and is well-presented, some important concerns should be clarified and revised before further consideration.

  1. The authors should highlight information about the databases used for collecting/extracting the data (for example, Web of Science, Scopus, Google Scholar,..) and what keywords were used during the literature search along with the period of studies included in the review. This ensures that the paper covers all available recent and relevant studies. All these points could be highlighted, at least, in the introduction section.
  2. In the Introduction section or in section (3. Flavonoids- structure, bioavaiability and molecular activity), I recommend the authors add additional information about other biological properties of flavonoids such as promising antivirals. I recommend the authors use the reference (DOI: 10.3390/v14030592).
  3. In figure 1, the structure of flavonoid is wrong; the displayed structure is a flavone strucure. I recommend the authors seek a help of Chemist to check the correctness of all chemical structures and other chemical aspects presented in the paper. 
  4. Finally, I recommend the authors double-check the whole manuscript for typing errors.

Author Response

Response to reviewer comments

All authors carefully read the reviewer's comments. All comments and suggestions have been included in the manuscript and the changes made are listed below:

1. A paragraph was added to the manuscript describing the method of searching for data using available databases of scientific publications, e.g. Web of Science or Google Scholar. The use of keywords was included and information was added to narrow the search results to the time period and selected, described flavonoids, which this article concerns.

2. In chapter 3 "Flavonoids- structure, bioavaiability and molecular activity"  there is a sentence (…) Therefore, the antioxidant activity of this group of compounds will have a beneficial effect in the comprehensive approach to the prevention and treatment of diseases with an immune, metabolic, viral and microbiological background (…). A reference proposed by the reviewer has been added to this sentence, which aptly complements the information on the antiviral and antimicrobial properties of flavonoids.

3. We agree with the comment of the reviewer, the structure presented in Figure 1 is a flavone backbone. In the current version of the manuscript, in Figure 1, the caption is "flavone backbone", which is consistent with the drawing. Other structural formulas of flavonoids were obtained from publicly available databases such as PubChem.
3. Spelling errors have been corrected and selected literary items in paragraphs with numerous citations have been corrected. The Zotero program and plug-in were used to prepare the references, which minimized errors in the numbering and titles of individual literature items.

Other changes to the original version of the article include comments from other reviewers.

Aleksandra Golonko

Reviewer 2 Report

Journal- International Journal of Molecular Sciences

Manuscript ID: ijms-2106803

Title-" Why dietary flavonoids have promising effect as enhancers of anthracyclines? Hydroxyl substituents, bioavailability and biological activity"

Reviewer comments

1.     Flavonoids are low molecular weight phenolic phytochemicals synthesized at particular locations in plants. The major groups of flavonoids are isoflavones, flavanols, isoflavane, flavanones, coumarins, chalcones, volatiles, and essential oils. Authors can also use the following recent review article (https://onlinelibrary.wiley.com/doi/full/10.1002/fft2.110), which describes the clinical application of flavonoids, their interactions and toxicities.

2.     Flavonoids are the primary active metabolites of plants, herbal medicine, dietary supplements, functional foods, etc. In interaction with cytochrome P-450 isoforms, they also act as agonists or antagonists with various nuclear receptors such as PXR, CAR and AhR and regulate their downstream genes, such as CYP3A4, CYP2C9 and P-gp transporters. Authors can add a separate section naming “flavonoids interaction with nuclear receptors (PXR, CAR and AhR)” and the following articles (https://doi.org/10.1039/C3FO60063G; https://doi.org/10.1016/j.phymed.2020.153416; https://doi.org/10.1016/j.heliyon.2020.e05357; https://doi.org/10.1016/j.jep.2022.115822; doi: 10.3390/molecules26082315; DOI: 10.1055/a-1557-2113; doi:10.3390/ijms21145025) should be helpful and can be used as references.

3.     Dietary flavonoids modulate the catalytic activity of cytochrome and transporter. In many cases, they increase the bioavailability of co-admisntarted drugs, including anthracyclines. This action is not always beneficial for patients. In many cases, it is clinically adverse due to over-accumulation of co-consumption drugs. Hence, the present manuscript needs to incorporate some clinical cases that occurred by flavonoids mediated over-accumulation of drugs as well as high clearance of drugs.

Decision- Minor Revision

Author Response

Response to reviewer comments

The authors carefully read the reviewer's comments. All suggestions were applied in the manuscript.

1. The review includes an article proposed by the reviewer concerning the ingredients of liquorice, which as a rich source of flavonoids, apart from many valuable, health-promoting properties, has a certain potential to increase the toxicity of commonly used drugs. The amendment was included in the last paragraph of chapter 3 "3. Flavonoids- structure, bioavaiability and molecular activity".

2. The dualistic effect of flavonoids as agonists and antagonists of nuclear receptors is distinguished in a separate subchapter 4.3. Flavonoids as agonist and antagonist of nuclear receptors”, which is a supplement to the described topic of the Nrf2/ARE pathway and refers to the activity of flavonoids as modulators of CYP isoforms, which was discussed in Chapter 5.

3. The most important clinical use of bioactive compounds of plant origin in combination with conventional chemotherapy was discussed in the "Conclusions".  The added paragraph describe the use of Aidi injection, which is a standardized mixture of plant extracts known in traditional Chinese medicine. This paragraph was summarized by the need to conduct more detailed studies on individual structures of flavonoid compounds in order to determine the relationship between the structure and their biological activity (QSAR/ SAR). So far, no clinical trials have been performed on patients undergoing anthracycline chemotherapy with simultaneous supplementation/injection of separate flavonoids, which makes it impossible to critically compare the effects observed so far in the numerous reports of in vitro studies. Our critical insights including limitations in bioavailability and therefore clinical applications of flavonoids are also briefly described in the „Conclusions”.

Other changes to the manuscript include those suggested by other reviewers.

Round 2

Reviewer 1 Report

The manuscript has been sufficiently improved.